# Neoliberal Rome—The Role of Tourism

**Roberta Gemmiti**

Department of Methods and Models for Economics, Territory and Finance, Sapienza University of Rome, 00161 Rome, Italy; roberta.gemmiti@uniroma1.it

**Abstract:** The primary objective of this paper is to analyze the main characteristics of recent tourism policies in Rome by describing the local modalities through which the neoliberal approach to urban strategies has been implemented. The first section highlights some general features of the city of Rome and its tourism, which are particularly useful for understanding the specificities of neoliberal tourism policies. The paper then proceeds to describe the most clearly defined neoliberal period of the city from 1993 to 2008, when the new Master Plan was drawn up to establish new policies and projects for tourism. The period that followed 2008 was marked by the gradual withdrawal of public action, which on the other hand has left ample freedom to the forces of tourism and globalization.

**Keywords:** neoliberal policies; Rome; urban tourism; center-periphery; Airbnb

## 1. Introduction

Neoliberalism is a powerful critical concept, which has gained much attention since the turn of the 21st century among scholars, journalists and politicians, and is rapidly becoming well-known and largely debated within the social sciences (e.g., Brenner and Theodore 2002; Larner 2003; Castree 2010; Jessop 2013; Springer et al. 2016).

According to Harvey (2005) well-known definition, *neoliberalism* is a theory of political economy which claims "human wellbeing can be advanced by liberating individual entrepreneurship and skills within an institutional framework characterized by strong property rights, free market and free trade" (ibid., p. 2). More broadly, neoliberalism is a political project aimed at extending competitive market forces, consolidating a market-friendly constitution, and promoting individual freedom (Jessop 2012).

After nearly two decades of analysis, it is even now an inspiring idea, with scholars still arguing about its nature, qualities and meanings, and even its validity and helpfulness as an analytical object (Jessop 2016; Purcell 2016).

In this considerable scientific production, both in theoretical debates or in specific themes and case studies, the link that plainly exists between tourism and neoliberalism has been generally ignored. Phenomena such as privatization and the commercialization of common goods (nature, culture, heritage) for tourism purposes, the deregulation and reregulation of sector institutions at the regional and local level (Schilcher 2007), the creation of flanking mechanisms and self-sufficient communities in the promotion of different forms of social tourism (Higgins-Desbiolles 2006), the power of the discourse surrounding new projects and the creation of new markets (Kline and Slocum 2015; Mosedale 2016) are all examples of a relationship between tourism and neoliberalism that is still not sufficiently studied and is largely confined to the sector literature.

The same thing happens in the specific case or urban neoliberalism, with tourism promotion representing one of the main dimensions of the translation of the free market logics into the field of urban socio-spatial relations (Rossi and Vanolo 2015); nevertheless, there are very few papers that explicitly address tourism policies in entrepreneurial urban governance, or regarding broad strategies for relaunching urban economic growth and competitiveness through the tourism sector (Fainstein and Gladstone 1999; Hall and Page 2006; Ioannides and Petridou 2016).

This paper aims at reinforcing the evidence of the role that tourism has played in the development of urban neoliberal policies.

The aim is to discuss the characteristics of neoliberal policies in Rome as an outcome of the city's institutional and spatial peculiarities. The paper also aims to explore how the specificity of tourism sector, as one of the most relevant dimensions of urban neoliberal policies, influenced and changed the urban development path.

The paper is organized into three parts. First, a description of the local context is given, focussing in particular on the peculiarities of Rome in order to better understand the unfolding of neoliberal forces in the city. The second part will describe the most relevant phase of local neoliberal policies, corresponding to the process of elaboration of the new Master Plan (MP), within which many urban projects related to tourism were considered as the keys to the revitalization of the city through major events, new architectural structures, as well as a creative and cultural offer. The third part analyses the period that followed the approval of the Master Plan, from 2008 up to today, focussing on the progressive retreat of local institutions and on the effects of the neoliberal stage becoming more and more evident.

## 2. Rome. Preliminary Reading Keys

Rome is world renowned for its cultural, historical, artistic and architectural heritage, which has always made it a desirable and almost obligatory destination for tourism. As the capital of Italy and Christianity, it is a political, administrative, military, diplomatic, cultural and cinematographic node. With more than 340 architectural and historical assets, and 188 archaeological sites it was declared a World Heritage Site by The United Nations Educational, Scientific and Cultural Organization (UNESCO) in 1980, and it is still one of the top fifteen cities in the world for international visitor arrivals, holding a remarkable share of the tourism market at international[1], national and local levels[2].

The relevance of tourism for the city of Rome explains its predominance in the strategies for relaunching the city from the beginning of the 1990s until the end of the 20th century, when increasing globalization made it imperative for capital cities to become competitive by attracting visitors, foreigners and workers. It was in fact during the period 1993 to 2008, under the left-wing leadership of two Mayors of Rome, first Francesco Rutelli and then Walter Veltroni, that the neoliberalism approach and a strong entrepreneurial style of government were particularly reflected in their strategies of urban planning and implemented tourism policies.

This approach, with a clear concern for tourism projects, had several characteristic features of the entrepreneurial city model (Harvey 1989): (a) an increase in the use of public/private partnerships, and the introduction of the logic and principle of economics into public choices and management; (b) a shift in the urban government agenda from a focus on public well-being and social justice to the aim of raising awareness of the urban image, in order to improve the city's competitiveness; (c) a political economy 'of place', with specific coalitions of properties development and financial operators aimed at 'selling' the best parts of the city.

The most apparent neoliberal period in Rome ended with the change of government in 2008. In the following years, the city witnessed a gradual and strong reduction of public action and commitment in the urban planning and daily management of the city, with more deregulated forms of tourist exploitation.

---

[1]  The fifth city in the world with a share of tourism GDP of 7.8, less than Paris (22.5) and London (15) but more than Barcelona (7.1) and Berlin (6) (WTTC World Travel & Tourism Council, fig. 3).

[2]  In 2017, the contribution of tourism to the city's GDP was around 5% and to the national GDP was 9%. The share of tourism GDP in the Italian economy is around 6%. The sector counted for 6.6% of the city's employment and more than 8% of Italy's national employment (WTTC World Travel & Tourism Council, p. 37).

Before going into a detailed description of the characteristics of these different phases of the city's government and its tourism policies, some preliminary keys for understanding the city of Rome can be briefly outlined.

As many have testified, Rome is an extraordinarily complex city; "a dynamic place, where the past is jumbled with the present, where people live amid a variegated landscape that is always changing. This is neither an austere nor a ruined city. It is a city of layers" (Agnew 1995, pp. 2–3); "a number of capitals in one" (Cellamare 2017b, p. 121). It is "a living miracle" . . . "a laboratory of intricate human relations and curious forms of sociability, of diffidence and civility, cynicism and humor, rudeness and kindness, a chaotic blend of distance and closeness, carelessness, apathy, and engagement . . . " (Clough Marinaro and Thomassen 2014, p. 1). In this complexity, both the horizontal network, those of the local relations between economic operators and institutions, and the vertical ones, through which the space has been transformed, are not at all easy to understand; much less so in a complex and contradictory city like Rome, which has been a stronghold of civilization as well as a symbol of deviance and corruption for centuries[3].

So, it is important to provide some preparatory knowledge about the city, which is relevant in order to understand how tourism (and its mutual historical relationship with the city), and the specificities of the local context have shaped the neoliberal project in Rome.

The first aspect to focus on is the role that land rents, land markets, and the construction sector have played in a city whose largely spontaneous and unregulated spatial development is due to the absence of a manufacturing sector and an enlightened entrepreneurial class capable of positively enhancing and orienting a city's transformation. The exploitation of the rent (land rent, tourism rent, building rent) has been at the base of the establishment of the "Urb's Regime" (the urban regime in Rome according to d'Albergo and Moini (2015)), a system of local power consisting of great landowners, entrepreneurs, banks, local and national politicians and administrators. This complex system is significant for explaining why neoliberal tourism policies have resulted in new buildings, new material facilities, new artistic and architectural constructions. Actually, the realization of renewal and redevelopment projects was a general trend for all cities worldwide from the end of the 1980s onwards; but in Rome this approach was particularly suitable for local entrepreneurs and institutions, whose city planning was based only on real estate sales and the construction of houses.

The second aspect concerns the deep gap between the central areas and the peripheral areas within the city, and between Rome and the surrounding City Councils or constituencies. These differences, which characterize Rome from the beginning of its recent development, involve aspects like the socio-economic conditions of local population and the distribution of economic activities and jobs, but also institutional relationships and the level of centralization of the decision-making process. In fact, Rome's City Council has always failed to recognize and to provide the non-central areas (within the municipality and around it) with more autonomy, and more financial assistance and resources which can allow them to develop their own path of independent development.

## 2.1. Urban Growth, Construction, Land Exploitation

Since the declaration of Rome as Italy's Capital in 1871, the city's growth was very rapid and largely spontaneous, disregarding the several Master Plans which tried to govern and regulate this

---

3　Rome has always been seen through good and bad stereotypes (Cellamare 2017b), and it has often considered a symbol of Italian attitude to malversation, the city where deviance and misappropriation's episodes usually occur. "Roma ladrona" (Rome, the thieving), the appellation given to the city by the Northern Political Party of Lega Nord, shows the national feeling in identifying the city with the lack of honesty in the public sector. Another example is the name of the most recent lawsuit, named Mafia Capitale (Capital's Mafia), which lasted from 2014 to 2017 and led to several local administrators and entrepreneurs being sentenced to jail for the illegal control of public contracts and economic activities.

process[4]. The unplanned development of the city was driven above all by land speculation and the growth of the building sector; this latter was the real engine of the city's economy in all the historical phases. Unfortunately, no plans succeeded in controlling Rome's spatial growth, but at least efforts were made to heal its damage, bringing services and infrastructure to areas where settlements had already risen abusively (Ferrarotti 1970). In some way, the institutions find convenience in the irregular unplanned growth of the city because this simultaneously solved the problem of employment and housing for all the migrants which continued to arrive in the Capital city, from the late Nineteenth century right up to the late Twentieth century, most of whom came from the surrounding regions and from the south of Italy (Insolera 1993).

The role of the land market and the building sector, the local institution's weakness and, moreover, the strong relationship between the economic-productive realm and the city's government (not always correct and legal), represent important elements in the analysis of Rome's tourism and the recent political choices for the sector.

In fact, the development of tourism is strictly connected to the city of Rome's growth, and the same institutional weakness that allowed the city to grow spontaneously, failed to organize and enhance tourism development. As a matter of fact, in the past, except for the management of major events, in particular the Holy Year in 1950 and the Olympic Games in 1960, in the case of Rome there was no serious policy for the organization and the promotion of tourism; more simply, the city had an evolution of tourism flows due to the extraordinary cultural resources and the concentration of many public and private functions of great national and international importance.

Spontaneity in tourism development was reflected in the localization of accommodation facilities that were mainly close to the historical city center and to the transport hubs, largely confining them to the 'core' of the Roman metropolis. In this way, many of the benefits from tourism did not go to the city (and still do not), but only to a few spaces and fell into a few hands, with the majority of the population of Rome receiving very little.

The fact is that Rome became the center of foreign arrivals (Floridia 1959) immediately after 1871 in a very sudden way, after several centuries in which it was one of the famous destinations for the few tourists of the Eighteen Century Gran Tour. The first period of tourism development in Rome established the main characteristics of the sector, in particular: the strong spatial concentration of accommodation within the city; the low quality of tourism and the low expenditure capacity of a good portion of the tourists; the dichotomy within the sector, if we consider the small number of historical and very prestigious hotels in the center and a large amount of medium and low-quality level accommodation structures in a few areas of the city.

In fact, since its definition as Italy's capital, Rome became the destination of a significant demand for housing coming from the employees and officials who moved there to give life to the bureaucratic apparatus of the State, and from the people who went there to look for new business opportunities. In those years the expansion of the city was extraordinarily fast: from 1871 to 1951 the population grew by eight times and actually doubled in the twenty-five year period from the mid-1920s to the 1950s. As a result, the city's residents in the mid-twentieth century were two fifths Roman and three-fifths born in another Italian city, region or foreign place (Floridia 1959).

Many people came to Rome for different reasons and often had a friend or a parent's place to reside in, or looked for somewhere cheap. So, until the early twentieth century, the local tourist accommodation grew above all with inns and pensions whose construction was part of the "building fever" that characterized that period (Girelli Bocci 2007). Most of the prestigious hotels had already been built in the 'trident' (that is the area which radiates from Piazza del Popolo and Piazza di Spagna and, through Via del Corso and Via del Babuino, right up to the Quirinale Hill). They hosted high

---

[4] Writing about the building expansion in Rome after 1871, Seronde Babonaux (1983) underlines the local government's effort to direct and to contain the growth of the city and she ascribes its failure to the pressures of national and local financial groups.

profile visitors such as politicians, the aristocracy, and international artists. In the meantime, the other types of tourist structures continued to grow in and around the city center, offering visitors (who did not stay with friends or relatives) average and low cost accommodation (Floridia 1959).

A new image of Rome that was more modern and similar to the Northern-European capitals came only after 1902 with international exhibitions. These types of events led to new hotels development, with the partial replacement of guest-houses and a general improvement of the quality of accommodation services.

Notwithstanding this, a very large part of the hotels and non-hotel facilities in Rome was owned by local families, as demonstrated by the historical reconstruction of the presence of joint stock companies in those decades in the city (Girelli Bocci 2007).

Even today, only a third of the prevailing juridical form of tourism agencies is constituted by joint stock companies, as for the large part it consists of individual enterprises, partnerships and other forms of hospitality (cooperatives, consortiums, etc.) (Camera di Commercio di Roma and ISNART  Istituto Nazionale Ricerca Turistica, tab. 16).

Furthermore, the process of globalization of tourism is rather limited in Rome, where the local market remains closed to international competition (d'Albergo et al. 2018). Most of the hotels in Rome that bear the mark of large international chains are owned by a few very influential local families. Often these proprietary families are more or less closely related to local builders, and this link, in a city where the land market is not only important from an economic point of view, but has often determined the city's style of development, has influenced and still influences public policy choices and characterizes their choices in tourism, helping to explain, as we will see, the way in which neoliberalism has been related, for a long time, to tourism policies.

### 2.2. Urban Governance and Centre-Periphery Gaps

The concentration of tourism in a few areas of the center, and in the hands of a few local entrepreneurs, is both the result of the city's process of development and one of the factors responsible for the center/periphery gap that has always characterized Rome, a gap that is primarily infrastructural, social and economic.

It is important to say that Rome is a very huge municipality. Its surface is 1287.35 square km, an area equal to the sum of eight large Italian cities, almost ten times the city of Milan. The resident population is 2,876,614 (31.12.2017). The functional region (the Italian Local Labor System) consists of a total area of over 6100 square Km, with more than 4,000,000 inhabitants (Figure 1).

The unplanned growth of the city has contributed on the one hand to entrenching a marked difference between on the one hand the densely populated and consolidated city of the central and semi-central areas, and on the other, the huge peripheral area that stretches from the center towards the Great Ring Road (Grande Raccordo Anulare—GRA) with the peripheral city growing around it and its outskirts.

The absence of any serious planning has sanctioned the future of the periphery, which has always lacked an adequate level of infrastructure and services; moreover, the presence of so many valuable functions in the center has naturally concentrated most of the resources and planning efforts dedicated to the whole city.

Rome is therefore composed of a few wealthy areas surrounding the historic center, including the northern and southern part of the consolidated city, and a very large portion, the one to the east and north, inside and outside the GRA, that is characterized by low-income levels, medium-low educational levels and low-skilled jobs (Alleva 2017; Celata and Lucciarini 2016; Monni et al. 2018).

From the point of view of activities, facilities, services and spatial distribution, Rome has remained a highly monocentric city, with over 70% of the employees of the entire functional region being concentrated in the center (ISTAT  Istituto Nazionale di Statistica). The densest and most central part of the city has a very strong polarizing effect; it is alive by day and at night it gives way to leisure activities and tourism (Lipizzi et al. 2018).

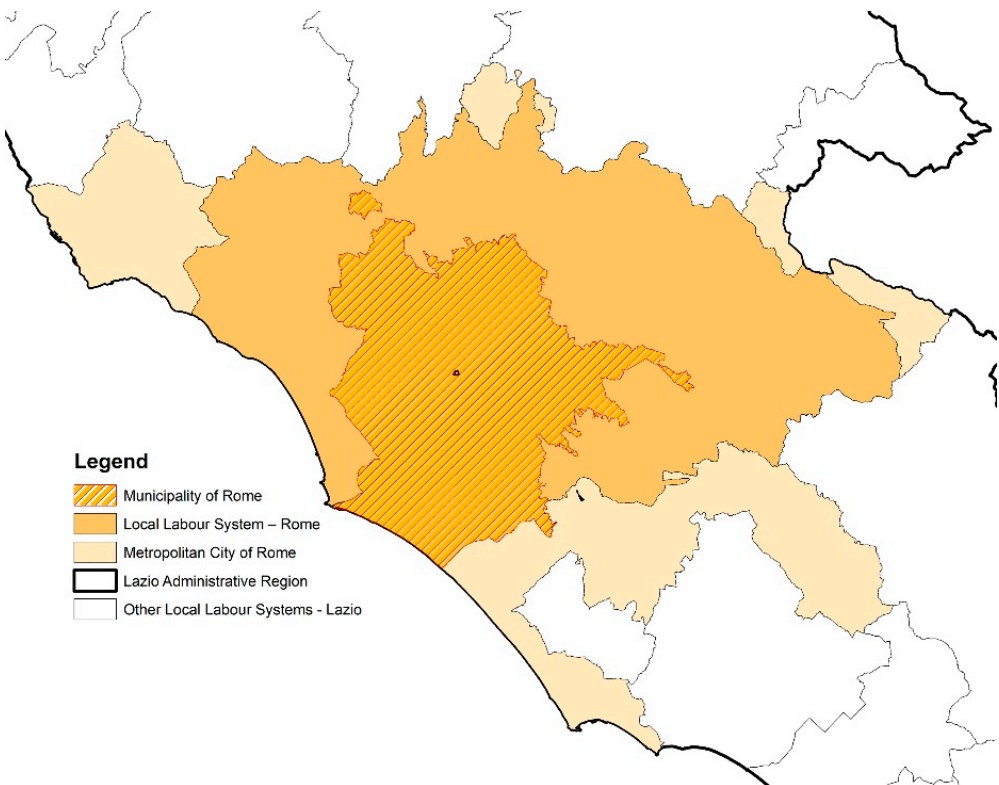

**Figure 1.** Administrative borders and Rome's Local Labor System. Source: https://www.istat.it.

The social and economic gap is, for the most, the result of the institutional weakness and the *lassaiz faire* attitude that has characterized the city's development in the past, but these policies are also the result of the traditional, centralized, and scarcely participatory management of power and resources. The city's governance has always been in the hands of the Mayor and the municipal bodies and, despite the hugeness of the municipality, which really requires a strong form of power decentralization, the government of Rome has not yet reached a truly metropolitan dimension (d'Albergo 2015).

Currently, the city is organized into fifteen sub-communal levels called Municipalities (named by numbers), each of which has demographic dimensions comparable to a medium-sized city in Italy (the biggest Municipality has around 308,000 inhabitants, while the smallest one has more than 131,000 inhabitants). Notwithstanding their demographic dimension, Municipalities have always had only very limited autonomy and few available resources.

Neither the relationships between the municipalities within the city nor the one between the city of Rome and the smaller cities of the province (in some cases really small) can be said to be truly democratic and participatory. The centralization of power and political choices, which are always made for the benefit of Rome's central areas, have produced a strong material and immaterial gap between the center and periphery.

The spontaneity of tourism development has made this sector partly responsible for this deep-seated gap. Since the beginning, Rome's center has polarized all the functions connected with the tourism sector. In 1881, the municipality of Rome received 76.2% of all the tourism arrivals of the Lazio region; and since then, these arrivals have always been concentrated in the central 'trident' with the traditionally old and prestigious hotels (Girelli Bocci 2006).

In the first decades of the twentieth century—when the first official statistics of the country became available—the polarizing capacity of Rome was even more marked: around 95.5% of the tourist arrivals registered in the province of Rome, and 91.1% of those in the entire Lazio region stayed in the city (Teodori 2006, p. 312).

Even after World War II, the spatial distribution of the tourism sector did not seem to change significantly in Rome's area, except for the expansion of guest-houses in the Prati neighborhood (close to Saint Peter's) and in the area around Termini Station (Colzi 2006). The development of campsite and extra-hotel accommodation only began outside the city center at the end of the 1950s due to two great events, the Holy Year in 1950 and the 1960 Olympic Games.

According to statistical data, in 1980, 78.5% of the hotel structures were still concentrated in Rome's historic center (the first Municipality), and 97.6% of these were found in the first, second, third and seventeenth Municipalities (Conti Puorger and Scarpelli 2006). It was only in the 1980s that there was a substantial growth in accommodation capacity in the outer parts of the city, although this occurred in a sprawled manner, with no spatial clusters or districts of particular relevance (Celata 2012). However, this new phenomenon did not affect the spatial distribution of the tourism sector in Rome, which even today still shows a concentration of more than 64% of the accommodation service located in the historic center, just 1.5% of the whole surface of Rome.

## 3. The Neoliberal Tourism Policies. The First Stage

The characteristics of the city, its deep internal differences, its urban regime and the tourism organization described above have influenced the ways in which the neoliberal approach shaped Rome's policies.

As already said in the introduction, the most significant phase of neoliberalism in Rome took place between 1993 and 2008, under the leadership of the Mayors Francesco Rutelli (1993–2001) and Walter Veltroni (2001–2008). Looking at what Brenner and Theodore (2002) outlined as the "destructive and creative moments of neoliberal localization", within their idea of actually existing neoliberalism in cities, it can be seen as Rome experienced several of these "moments" before and during the Nineties (idem, pp. 369–72): (a) the recalibration of intergovernmental relations, with an important process of devolution from central government to metropolitan areas and to Rome in particular as the capital of Italy; (b) the reconfiguration of institutional structure, with the introduction of new networked forms of local governance based upon public-private partnerships, through which the local élite was enabled to change the city in a neoliberal way; (c) restructuring strategies of territorial development, in particular the European spatial polycentric model that functioned as a planning framework for neoliberal policies in Rome; (d) transformations of the built environment and urban form, especially through mega-events strategy for urban growth and competitiveness; (e) re-representing the city, mobilizing entrepreneurial discourses and representations.

During Nineties, a dynamic climate and increasing trust in the aims and values of competitiveness and economic growth were quite diffused in Italy, as well as the pervasive use of flexible planning tools and public/private partnerships, both in the decision-making and implementation phases. This trend was heralded by a new law which allowed citizens to elect the Mayor (Law 81/1993), and all Italian cities followed suit (Semi 2017). Many urban projects for the development and regeneration of sites were also often connected to European funds and great events (e.g., the Colombiadi, the Olympic Games, the Expos). On the whole, there was a strong use of public-private partnerships, which reflected the tendency of cities to operate autonomously and self-determine their development paths, involving also a greater diffusion of small projects, often aimed at achieving the European principles of cohesion, sustainability, innovation and, in the case of Rome, spatial and functional polycentrism.

This general trend was boosted in Rome by the greater autonomy and powers designated to the city by the new status of 'Capitale dello Stato' (Capital of Italy[5]), a new special funds program for Rome Capital (1990), and many specific funds for the organization of Jubilee 2000.

---

[5] This new status was promoted through different stages. Starting from the first change in the Italian Constitution (2001), law 42/2009 gave Rome a new and more autonomous status in the fields of administration, funds and laws; the implementing decrees gave the city a new name (Roma Capitale), a new institutional structure, new functions and more power to the Mayor. The last important stage was the new status of Rome as a "Metropolitan City" (L. 56/2014).

All these funds, projects and targets converged and found coherence in the new Master Plan for Rome (MP), the main and most important outcome of the two left liberal City Council planning boards. The Master plan proposed an ambitious strategy for the transformation of the city. The aim of the system of power (the Urb's Regime) was to relaunch Rome's economy and international competitiveness through a radical change of its spatial organization and a more balanced distribution of functions and economic activities. Tourism was meant to be central in the plan, as it was considered to be both a key economic sector and a set of material and immaterial 'tools' for promoting the visibility and competitiveness of the metropolitan system imagined for the future (Ciccarelli et al. 2012; Gemmiti 2008).

The plan divided the municipal area into three different zones, each with three equally different planned policies (Figure 2): (a) the extra-urban area, (b) the consolidated city, (c) the city yet to be restructured and transformed. The extra-urban area consisted mainly of natural parks aimed at nature conservation and the promotion of agriculture and sustainable activities; the consolidated city was meant to enlarge the portion of the city which needed to be safeguarded, so the central area of Rome was substantially extended well beyond the most ancient zone; the city to be completed and restructured was the huge remaining part where the large part of the new forms of land speculation took place[6].

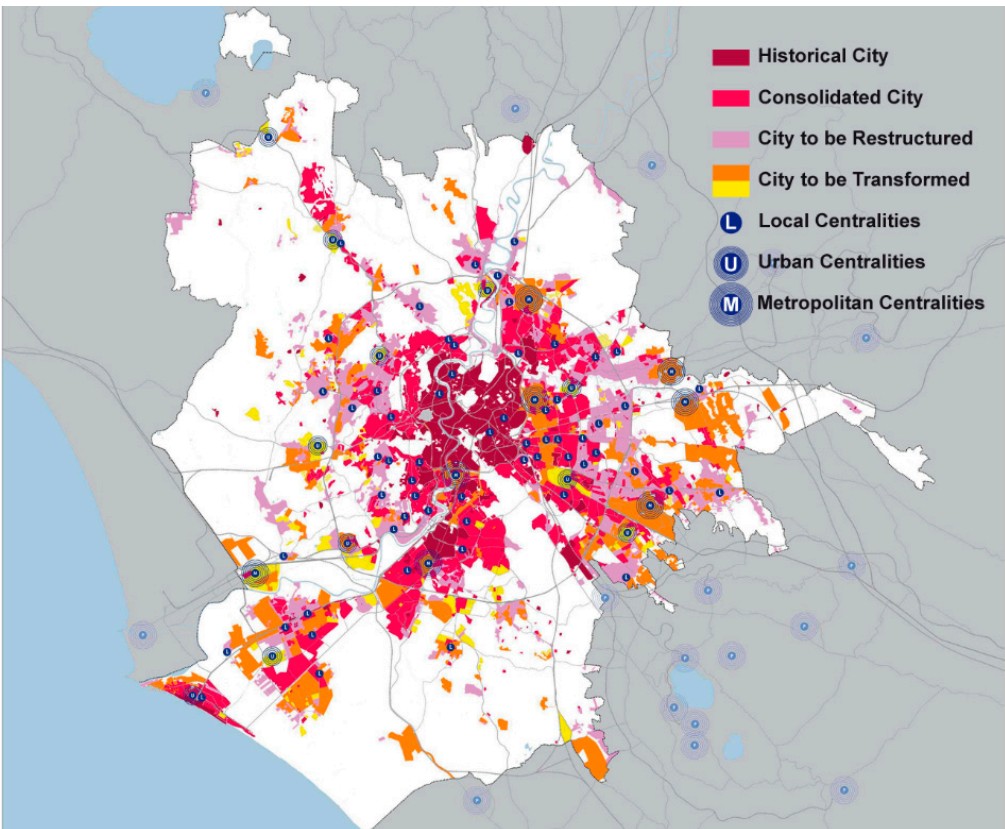

**Figure 2.** The main features of Rome's Master Plan. Source: http://www.urbanistica.comune.roma.it/prg.html.

In this spatial subdivision, the plan envisaged the constitution of a certain number of functional nodes, the so-called 'Centralities' (Figure 2), a system of functional nodes which are called to bring more

---

[6]    The city to be "restructured" recalls the famous label "Restructuring" used in Eighties in order to describe the new political project for cities. The term suggested "a sequence of breaking down and building up again, deconstruction and attempted reconstitution" (Soja 1987, in Brenner and Theodore 2005, p. 101). During the 2000s, the term became complemented by the concept of neoliberalism.

polycentrism and equilibrium within the local metropolitan area. These Centralities were organized according to three hierarchical levels, depending on the quantity and quality of the dimension of the project and the importance of the functions that were intended to be located: the Local Centralities (more than 60), the Urban Centralities (10), the Metropolitan Centralities (8). The local centralities consisted in punctual and ordinary interventions at the neighborhood level and are not relevant to the present aim; actually, the urban and, moreover, the metropolitan central places were meant to act as "structuring projects", as real propulsive centers of transformation and development for the city. On these functional nodes rested the overall urban strategy (Busti 2018), and it is in these nodes that we may seek the role of tourism in the general neoliberal development strategy planned for Rome.

The metropolitan centralities were realized close to the GRA and outside the GRA, and most of them are on the right side of the Tiber river. The plans forecasted a very large set of town facilities to be built in this area, from new mega-hotels, impressive architectural structures, huge spaces for trade and commerce[7], many suburban areas for new residents. The most relevant interventions for promoting tourism and Rome's attractiveness were localized in a very small corridor going from the city center, following the Tiber river, to the coast and Fiumicino airport (Figure 3). Here, the neoliberal policies for tourism planned several interventions in order to enhance the city and its cultural prominence, creating a dynamic business atmosphere and economic vitality.

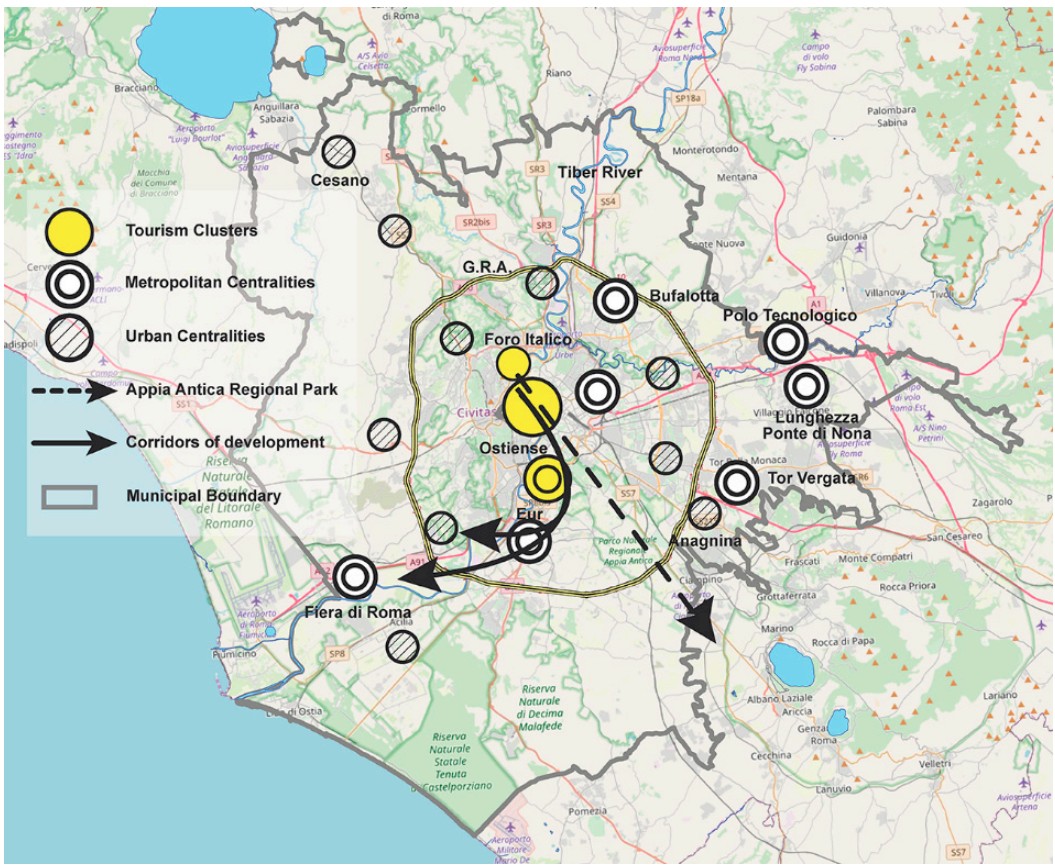

**Figure 3.** Spaces of Rome's development and tourism clusters. A synthesis. Source: own elaboration.

Many projects for urban redevelopment and regeneration have been realized in the three small areas identified in the Figure 3 as tourism clusters. According to the Mayor's statements, the aim was not so much to increase the number of visitors but rather to extend the average stay of tourists, bringing

---

7   In particular, shopping malls that were developed in Rome largely in the 1990s.

Rome to be like London and Paris. So, through significant new architectural symbols, often projected by international 'Archistars', the plan aimed to design new and different resources and spatial clusters for tourism. The first cluster included the Flaminio quarter and the Foro Italico stadium and sports complex which hosted the 2009 Swimming World Championship; this cluster was actually arranged through several facilities that improved the existing provisions since Fascist era and, more recently, the Olympic Games of 1960[8]. In the Flaminio/Foro Italico area, has been built a new bridge (Music Bridge) crossing the Tiber river and linking the sport area to the cultural 'core' consisting of the new music Auditorium by Renzo Piano and the new Zaha Adid' MAXXI Museum. This first cluster is very close to the historical city center and Piazza del Popolo, the cultural heritage core in Rome. From here, the corridor of the new neoliberal tourism strategy included the renovated Centrality of Ostiense-Old (Shambles) Market that has been designed to be a multifunctional cultural and tourism cluster, particularly aimed at young people (a new sort of Covent Garden) and at promoting new expressions of culture in modern day entertainment and the performing arts. The project also forecasted the construction of a new University area[9], a Science Town and a Science Bridge, the new Contemporary Art Museum, the City of Taste, a new Media Library, a Municipal Museum, and many other small initiatives.

Going to the south of the city, tourist-cultural development included the EUR neighborhood which, since its construction in the 1930s, represents the modern face of the city[10]. This area was still at the center of a remarkable regeneration project[11]. In this area, defined as the new Rome Business District, cultural tourism was intended to pave the way for business tourism, which included the huge strip of space reaching the sea and Fiumicino Airport. Here the plan consisted of a new Trade Fair of about 186,000 square meters, together with large residential areas and a big shopping mall. In truth several accommodation facilities have also been projected, as well as a skyscraper of almost a hundred meters to mark the 'gate' to Rome from the Airport.

The entire complex strategy of promoting the city's economy and competitiveness was meant to be realized through a few functional nodes, in a narrow spatial area which connected the center to the main airport. Tourism was supposed to be one of the most relevant economic engines for relaunching the city, but the neoliberal approach that inspired the plan forgot many of the relevant resources that the city offers. The most important is the huge area around the ancient Roman road the Appia Antica, which includes a regional park and is one of the most important and significant archaeological areas in Rome. Despite the extraordinary cultural importance and the numerous projects that many experts and scholars have tried to define over the decades, such as conservation regulations to stop the illegal buildings, Appia Antica still lacks the much needed interventions for its protection and promotion as an extraordinary cultural heritage site.

## 4. The Second Stage. The Retreat of the Public

The above described neoliberal period was characterized by the great confidence in the possibility of making Rome a truly global city. Local institutions called this planning approach the 'Rome Model', in the belief that specific interventions could act as a development model. Broadly, in that period the main indicators on a large geographical scale were quite good: the population grew by 2.3% from

---

[8]　The Foro Italico area was actually a fall-back after the failure of the planned sports center in the Centrality of Romanina, in the eastern area of the city. There, it is still visible from any point of the city a huge abandoned structure of what should be the new swimming facility projected and never completed by Santiago Calatrava.

[9]　The third public university in Rome was established here—the University of 'Roma Tre'.

[10]　In fact, EUR is the acronym for Esposizione Universale Roma (Roman Universal Expo). It is expected to function as the modern face of the city because is normally crossed by travelers coming from the airport and going to the historic center.

[11]　Several interventions were planned in EUR neighborhood, some of which have been only partly completed. For example, the project which planned to substitute the two old towers (Torri delle Finanze) built during 1950s by Cesare Ligini, with two new towers (never realized). The old towers have been partially destroyed, notwithstanding the remarkable architectural importance, and left to rot.

1995 to 2005[12]; the per capita GDP increased in relation to the EU-25[13]; the employment rate went up from 49.57% in 1995 to 59.83% in 2005[14]; the economic improvements seemed to follow the Lisbon recommendations too, e.g., judging by the female employment growth rate which rose from 34.87% in 1995 to 51.13% in 2005, and judging by the reduction in long-time unemployment from 4.2% in 1995 to 2% in 2005 in relation to the working population. The favorable climate and the strong investments in the tourism sector started attracting new activities, such as the Congress Tourism Exchange hosted in the new trade center called the 'Fiera di Roma'.

Nevertheless, the literature highlighted many controversial aspects emerging from the city's governance in those years, in particular the prominence given to private actors in the decision–making and, most of all, the strong connection established between the construction sector and spatial planning choices (Busti 2018). In fact, the institutions did not share the decision-making process with citizens, towards whom dialogue and openness to participation was rather superficial and unauthentic[15]; the urban regime forced them to interact with entrepreneurs, in particular, local entrepreneurs, notwithstanding that the political discourse emphasized the importance of capturing international capitals. The 'Rome Model' was represented as a great collective effort to modernize the city but, indeed, it was a strategy that drew on global logic and processes (urban image improvement, place marketing, creative planning, built environments, etc.) for very local advantages. Only a small part of the city was involved and a gap was created between the center (protected and enhanced) and the periphery (built and exploited), producing a sense of exclusion for local people (Berdini 2000).

The neoliberal approach, moreover, precluded the possibility of dealing with the very problems of the city: poverty and the struggles of a large part of the population, the housing market and the lack of affordable houses, the inadequacy of infrastructure involving transport, energy and waste, the problem of the inclusion of ethnic minority groups and immigrants (AA.VV. 2007; Berdini 2008). There was nothing for the peripheral areas, no new services and tourism facilities, but instead new and expensive houses and spaces for commerce and shopping malls (Cellamare 2017a; Erbani 2013).

Probably, the detachment between the people of Rome and the left-wing parties responsible for the neoliberal policies in the fifteen years' period, helped pave the way for the succeeding center-right council led by Gianni Alemanno (2008–2013). This was followed by the center-left council led by Ignazio Marino (2013–2015) and, then, in June 2016 by the election of the current Mayor Virginia Raggi.

The neoliberalism was still active in the first stages of Gianni Alemanno's policies, when some projects already planned by the previous council were proposed, developed further and implemented. The most remarkable new project regarded the realization of a second 'tourism pole' in the south of the city, as an alternative to Rome's historical one. In fact, several tourism projects were designed in the south of Rome, including a theme park based on Ancient Rome close to Fiumicino Airport a new Formula 1 circuit in EUR (these latter two projects have never been built or even begun) and a new Aquarium under the artificial lake in the EUR district (recently completed).

However, the corruption which seems to have characterized Alemanno's leadership to date[16] prevents an objective analysis of the tourism policies which were carried out under his governance as Mayor. For sure, the neoliberal period ended in 2008, leaving the field to a progressive retreat of institutional public diligence.

---

[12] Rome NUTs 3. ISTAT data, www.demo.istat.it.

[13] In Rome (NUTs 3) set the GDP per capita at 100 in 1995, in 2005 it raises to 102.37 (values in Power Purchasing Parity). In absolute terms, according to EUROSTAT data, GDP at market price was 68,824.70 million euros in 1995 and 119,757.10 in 2005; in real terms, it rose from 78,209.18 million euros in 1995 to 111,494.48 in 2005.

[14] ISTAT, Survey on Manpower, miscellaneous dates.

[15] It was significant, in 2008, the establishment by the population living in the Centralities of a "Network of Urban and Metropolitan Centralities", aimed at asking the realization of what was agreed during the participation to the planning process (http://centrumroma.blogspot.com/).

[16] Gianni Alemanno was sentenced at first instance in February 2019 for corruption and illicit financing within the "Mafia Capitale" lawsuit.

This drop in public activities for the city's development was due to several processes, most of which happened at the national level. Today, like in the past, the city is suffering from problems and processes that are not only local but even linked to the national level. The first problem is that of the lack of a strong and transparent leadership. Unfortunately, the recent crisis in Italian politics has produced an incompetent leading class, even for the Capital City. The second is the dramatic reduction in local funding that has hit all Italian municipalities since 2010, which was the peak of the world financial crisis, in the national effort to reduce the public deficit. Finally, the difficulties in the institutional dialogue between the leadership of Rome Capital and central government, for which the investment capacity of the city of Rome was objectively reduced to less than a quarter compared to what it was in the years before 2008 (Causi 2017), and it is still now less than the other cities of the country.

As regards the leadership of Mayor Ignazio Marino, from 2013 to 2015, he gave great prominence to tourism in his speeches but, in actual fact, his policies were far from effective and proved to be quite poor in their impact. He focused exclusively on a few specific interventions in the historic center, without long term vision. His very first act was the closure of the Via dei Fori Imperiali (from Piazza Venezia to the Colosseum) to traffic, and this can be considered, together with some measures for the urban decor in Rome center, the most significant action taken for the promotion of tourism.

The current City Council planning board, led by Virginia Raggi, can be considered even less significant in the field of tourism. As now known, the present board is having enormous difficulties in managing the city's basic needs, so it is difficult to say if and when a strategic vision of urban and tourism development for Rome will be available.

In the absence of any kind of policies or visions for tourism, the passage from neoliberalism to a completely unregulated activities are taking place in Rome, with really small benefits for local people. It is a new modality of the self-made city (Cellamare 2014) that is similar to the one encountered in the decades of Rome's spontaneous growth during the twentieth century.

One example of this type of unregulated activities is provided by the fast growth in Rome of networked hospitality businesses. For example, in 2015 the Airbnb network gained 7.1% of all the houses in Rome's city center, and this number rose to 8% in 2016 (Picascia et al. 2017). It is an impressive amount if we consider the small surface of Rome's historical area.

The effects that these forms of accommodation produce on cities and their social and economic structure are quite clear and seem to reinforce traditional and well known urban inequalities and local conflicts. The literature focus on the phenomenon of the centralization of housing supply in the hands of a few large operators, with a great impact on housing market and the economic benefits of tourism concerning the city and its inhabitants only in a very marginal way (Blanco-Romero et al. 2018; Capineri et al. 2018; Yrigoy 2018); moreover, the centralization of the hospitality business also has a very strong spatial dimension, in areas where the presence of traditional accommodation and B&Bs is already very important (Garcia-Ayllon 2018). These and other phenomena could lead to a worsening of the gentrification processes in a city's center and more and more inequity in the injustice of urban development (Còcola Gant 2016; Horn and Merante 2017).

Many of these processes are already happening in Rome. According to a recent study on this topic, 4 of the 155 urban areas of Rome host 35% of the housing and these four areas receive 58% of the income; the bed concentration index in the city is only slightly lower for Airbnb than that of traditional hotel accommodation; the center/periphery price pattern penalizes those who reside outside the traditional areas; the Airbnb system helps to strengthen the process of gentrification and displacement of residents and services to the city, emptying and disenchanting what was once the vital center of Rome and its population (Celata 2017).

## 5. Conclusions

This paper has described the recent planning experience in Rome, with the aim of debating the role of tourism and the local context in the unfolding of a neoliberal political project within urban areas.

The idea was to discuss how neoliberalism has taken specific features according to (a) the distinctive spatial and institutional arrangement of a city as complex and unique as Rome is and (b) the nature of the tourism sector, in as much as it has been central in the relaunch of the city's image, competitiveness, form and structure.

As was previously detailed, the neoliberal project was particularly evident in Rome between 1993 and 2008, coinciding with the process of elaboration of the new Master Plan for the city.

In that period Rome experienced many of the destructive and creative moments of what Brenner and Theodore (2002) defined as "actually existing neoliberalism" in cities: a greater level of autonomy and power, following a strong process of devolution from central government to the lower levels; the opportunity to use new forms of local governance based upon public-private partnership, through which the local élite worked to change the city; a new model of territorial development that was subsequent to the European guidelines for a more polycentric spatial structure in the European Union which were used as a framework for transforming the built environment and the urban form; a new way of representing the city, with entrepreneurial discourses and representations aimed at improving the competitiveness level and the economic strength of the city. In this neoliberal stage in Rome's governance, the tourism sector played a central role.

As the paper has demonstrated, several specificities of the city influenced the nature of the neoliberal policies implemented in Rome, and in particular: the institutions, the system of power described by d'Albergo and Moini (2015) like an "urban regime", which has always linked together local and national politicians, banks, great landowner and local entrepreneurs in the building sector and tourism; the productive structure, a local economy based on services, tourism and the public sector, which has been protected by the roman élite from globalization; the unbalances in the city's social and spatial organization, and particularly the deep gap between the central areas of Rome and all the huge spontaneous and often abandoned peripheral areas. These characteristics shaped a set of neoliberal policies developed by a few for a few, sharpening the internal divisions, the economic weakness, the social problems, and the structural inadequacy of the city.

If the consequences of the neoliberal approach in Rome can be largely discussed through the peculiarities of the city, a significant role has also been played by the tourism sector's spatial and economic logics. In the functional nodes implemented by the Master Plan with the aim of giving a new competitive and more balanced spatial organization to Rome, tourism resources and facilities, material and immaterial structures and events are always included. These structures added up to the already complex and divided city, with the strong spatial concentration of tourism promotion in a limited part of the city reinforcing the dramatic spatial and social gaps. Moreover, the non-central spaces have been used to build structures or amenities useful for the relaunch of the city center which resulted completely detached from the peripheral areas' needs and capacity of development.

Not only did the distortion of the urban regime and the attitude to speculation and land exploitation shape the neoliberal policies in Rome, but even the logic and functioning of tourism contributed to the support of a completely wrong idea of development for the city.

Tourism focuses on promoting the best part of the city and this often leads to neglect of the current daily problems of the city and its inhabitants. The danger connected to the free market forces of the tourism sector is evident in the second stage described in the paper.

Since 2008, it has no longer been possible to speak properly of a neoliberalism planning and policies, particularly regarding the general strategies for urban development and tourism, even though the mayor Gianni Alemanno and his Board completed some structures included in the MP and tried to propose some new mega-projects (which were never implemented). The progressive retreat of public hands from the governance of the city started with Mayor Ignazio Marino and became fully visible with the current Mayor Virginia Raggi, allowing tourism to freely exploit the most relevant resources of the city, leaving Rome and its inhabitants with almost nothing. Spontaneity and the *lasseiz faire* form of populism have led Rome towards new and more dangerous forms of colonization and inequality.

**Funding:** This research was funded by Sapienza-University of Rome (grant number: 000041_17_RDB_Ateneo2016_Gemmiti).

**Conflicts of Interest:** The author declares no conflicts of interest.

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
