# Peer review of "Neoliberal Rome—The Role of Tourism"

_socsci, doi:10.3390/socsci8060196_

Round 1

Reviewer 1 Report

The article describes tourist policies developed by the Municipality of Rome in the period 1993 to present, within the framework of the neoliberal urban. The topic is interesting and is properly developed. The text collects the main recent contributions on the development of the city and use them to describe the framework of policies developed in recent years. The approach, however, is overly descriptive, and the analysis use mostly secondary sources. The text would greatly improve if the policies described be conceived in a more complete analysis of the conceptual framework, and if the author will provide primary information to assess the impact on the territorial and social model of the city.

Author Response

First reviewer

The article describes tourist policies developed by the Municipality of Rome in the period 1993 to present, within the framework of the neoliberal urban. The topic is interesting and is properly developed. The text collects the main recent contributions on the development of the city and use them to describe the framework of policies developed in recent years. The approach, however, is overly descriptive, and the analysis use mostly secondary sources. The text would greatly improve if the policies described be conceived in a more complete analysis of the conceptual framework, and if the author will provide primary information to assess the impact on the territorial and social model of the city.

Response: Thank you for the suggestions, that made me think deeply about the possible improvements of the paper. I think to have enhanced the problem of the conceptual framework, re-writing completely the introduction and the conclusion sections, and including several thoughts and references in the main text. In my opinion, the paper Is now substantially less descriptive and more aimed at discussing the case study in a theoretical perspective.
About the primary information and the assess of the impact of the policy on the city, I’m sorry I couldn’t do anything of the kind. I’m sure the paper would greatly improve with such a work, but for me this means to do a completely different research path, something similar to write a new paper. Moreover, you have no idea how hard is to obtain primary information and statistical data about Rome and in particular about tourism sector. However, I appreciate the suggestion. It’s a good idea and I’d like to follow it in the next future.

Reviewer 2 Report

This study is an interesting analysis of the policies that have shaped the cultural heritage tourism sector of Rome, Italy. It focuses on the Mayoral and Planning councils’ neo-liberal techniques to help expand the market/availability of resources for tourists with the goal of making Rome a top global tourist destination. It analyzes the historic trends in tourism in Rome from the period of the Grand Tour of the 19th century up to today, with special emphasis on the 1990s and early 2000s.

The overall content of the article is good, and it is well written; however, there are some issues that need to be addressed to increase the significance of the content and interest to a wide range of readers. First there needs to be a significant expansion in the literature regarding the term Neo-Liberalism. Such expansion is needed to not only describe the general political and economic ideals of the term, but also better situate how such policies (both positive and negative) translate from the general theoretical sense to the physical manifestation in Rome’s tourism sector (a grounding of the theory, if you will). Included in a list below are some other thoughts/issues related to the article.

Lines 1-80 read like a tourist pamphlet or a tourism website.

Data on international dollar value of tourism, %GDP, etc would be helpful not only for Rome, but also at the national level since major tourism driver. Pre 1990 tourism levels would be helpful to establish % change and adding significance to the impact of the neo-liberal transformation of the tourism sector.

Citing Agnew about Rome is strange/dated inclusion. Maybe more recent sources, including some Italian authors would be better.

Source and more description needed for deviance and corruption statement on line 109.

An awkward switch to first person starting on line 110.

161 should say Grand Tour—include a brief discussion of this idea plus some sources either from the period from travelers who visited Rome, or from modern scholars interpreting that period

An interesting aspect that is missing from the discussion of historic tourism in Rome (lines 113-210) is the Fascist Period wherein Italian, specifically Roman, identity came to the forefront of national(istic) identity and what impacts this had not only on tourism, but heritage site development. The period is briefly mentioned on line 338.

 A map of Rome and its FUA would be helpful in section 2.2.

Figure 1 needs revisions. The scale is too small to see details (such as the centralities distinctions) and the nuances between the colors (burgundy and red and orange and yellow).

 More discussion is needed concerning the terms Urban and Municipal centralities (in relation to the Tourism clusters) in the text because they feature prominently on the figures.

Maybe include sources/stories that further develop the idea that citizen input was superficial (line 378)

The discussion of AirBnB encroaching on the private housing stock is interesting (432 ff) and reminds me of the most recent UN study on housing issues/availability/inequality (especially the encroachment of rental units and AirBnB) at a global level.

 The conclusions are a bit abrupt. More discussion is needed concerning the decline of global tourism during the global financial collapse, but also regarding the fact that is has been nearly a decade since that period. It would be interesting to see if there are any recent rebounds and/or policy changes to bring tourists back to Rome.

Author Response

Second Reviewer 

The overall content of the article is good, and it is well written; however, there are some issues that need to be addressed to increase the significance of the content and interest to a wide range of readers. First there needs to be a significant expansion in the literature regarding the term Neo-Liberalism. Such expansion is needed to not only describe the general political and economic ideals of the term, but also better situate how such policies (both positive and negative) translate from the general theoretical sense to the physical manifestation in Rome’s tourism sector (a grounding of the theory, if you will). Included in a list below are some other thoughts/issues related to the article

Response: I’d like to thank the referee for the suggestions, which helped me really much in improving  the paper. I did try to “ground” the theory to the case study, changing completely the introduction and the conclusion, and including several thoughts and references within the main text. I’m quite satisfying, even considering the short time. I included or erased several things, according to the suggestions. Below the list. Only two things remain to do (points 5 and 6); I hope to do it in the next round. Thank you very much

1.      Lines 1-80 read like a tourist pamphlet or a tourism website.

Response: True. I cut nearly everything off

2.      Citing Agnew about Rome is strange/dated inclusion. Maybe more recent sources, including some Italian authors would be better.

ResponseThe inclusion of Agnew is due to the idea of “layers”, that is particularly suggestive for Rome. There are many other possible sources, but not as much significant. However, I added one more italian author talking about the great complexity of the city.

3.      Source and more description needed for deviance and corruption statement on line 109.

Response: I included a reference to the national feeling about Rome as a place where politic's corruption and malversation normally happens. I mentioned the recent trial named 'Mafia Capitale' as an example of the link between the city and the malversation.

4.      An awkward switch to first person starting on line 110

Response: Changed

5.      161 should say Grand Tour—include a brief discussion of this idea plus some sources either from the period from travellers who visited Rome, or from modern scholars interpreting that period.

a.

Response: I think a note about this topic will help. I couldn’t do it now because the time was too short  

6.      An interesting aspect that is missing from the discussion of historic tourism in Rome (lines 113-210) is the Fascist Period where in Italian, specifically Roman, identity came to the forefront of national(istic) identity and what impacts this had not only on tourism, but heritage site development. The period is briefly mentioned on line 338

Response: I think this is not a very relevant point. I may include a note (I couldn’t do it now for the short time)

7.      A map of Rome and its FUA would be helpful in section 2.2.

Response: I added a map with the administrative borders and the Local Labour System of the city

8.      Figure 1 needs revisions. The scale is too small to see details (such as the centralities distinctions) and the nuances between the colors (burgundy and red and orange and yellow).

Response: Figure 1 (now is figure 2 in the revised manuscript) shows a map by Rome's Planning Office. It is the picture with the best resolution available online. Unfortunately, it is impossible to change it without the original file. I've asked to the Planning Office for a different map and I’m still waiting for a response. However, what is important in the picture is the size and amplitude of the three areas, and in particular the city to be “protected” (the central part) and the city to be transformed where the bold land exploitation is planned. The figure can be erase, in any case, because the contents are not crucial for the paper.

9.      More discussion is needed concerning the terms Urban and Municipal centralities (in relation to the tourism clusters) in the text because they feature prominently on the figures.

Response:  Actually, the difference between Urban and Metropolitan centralities is clear in the Plan but not in the projects they really realized over time. This is why I can’t detail the contents in either of them. I added a note about Foro Italico sport-events cluster.

10.   Maybe include sources/stories that further develop the idea that citizen input was superficial.

Response:   Good. I added a note about it.

11.   The discussion of AirBnB encroaching on the private housing stock…..

Response: I added some more references in the critical discussion.

10.  The conclusions are a bit abrupt. More discussion is needed concerning the decline of global tourism during the global financial collapse, but also regarding the fact that is has been nearly a decade since that period. It would be interesting to see if there are any recent rebounds and/or policy changes to bring tourists back to Rome.

Response : I completely re-wrote the conclusion. Not in the sense suggested because, broadly speaking, there are no policies at the moment in any sector of the city public life. Tourism is completely abandoned as it is the whole city with this Mayor and her Board.

What I actually tried to do it’s a better link between the conclusion and a) the aims of the paper b) the topic of tourism/urban neoliberalism/Rome

Round 2

Reviewer 2 Report

The changes made throughout this second version make the paper much more readable. The elaboration of the idea of neoliberalism and the inclusion of more sources related to Rome are good additions, as is the map of the city [figure 1] (maybe it could use more layers such as roads or location of the city center/Roman Period city as tourists and visitors would be able to relate to those rather than the general polygons presented here). Some minor revisions to the recent changes related to word use/grammar is needed, but it is generally well written.